# Abdominal Obesity-Related Disturbance of Insulin Sensitivity Is Associated with CD8^+^ EMRA Cells in the Elderly

**DOI:** 10.3390/cells10050998

**Published:** 2021-04-23

**Authors:** Tim K. Boßlau, Paulina Wasserfurth, Britta Krüger, Thomas Reichel, Jana Palmowski, Josefine Nebl, Christopher Weyh, Alexander Schenk, Niklas Joisten, Frank Stahl, Stefanie Thoms, Kristina Gebhardt, Andreas Hahn, Karsten Krüger

**Affiliations:** 1Department of Exercise Physiology and Sports Therapy, Institute of Sports Science, Justus-Liebig-University Giessen, Kugelberg 62, 35394 Giessen, Germany; tim.k.bosslau@med.uni-giessen.de (T.K.B.); thomas.reichel@sport.uni-giessen.de (T.R.); jana.palmowski@sport.uni-giessen.de (J.P.); christopher.weyh@sport.uni-giessen.de (C.W.); kristina.gebhardt@sport.uni-giessen.de (K.G.); 2Institute of Food Science and Human Nutrition, Leibniz University Hannover, Am Kleinen Felde 30, 30159 Hannover, Germany; wasserfurth@nutrition.uni-hannover.de (P.W.); nebl@nutrition.uni-hannover.de (J.N.); hahn@nutrition.uni-hannover.de (A.H.); 3Nemolab, Institute of Sports Science, Justus-Liebig-University Giessen, Kugelberg 62, 35394 Giessen, Germany; britta.krueger@sport.uni-giessen.de; 4Department of Performance and Health, Institute of Sports and Sport Science, Technical University Dortmund, Otto-Hahn-Straße 3, 44227 Dortmund, Germany; alexander.schenk@tu-dortmund.de (A.S.); niklas.joisten@tu-dortmund.de (N.J.); 5Institute of Technical Chemistry, Leibniz University Hannover, Callinstrasse 5, 30167 Hannover, Germany; stahl@iftc.uni-hannover.de (F.S.); thoms@iftc.uni-hannover.de (S.T.)

**Keywords:** elderly, obesity, T-EMRA cells, kynurenine pathway, insulin resistance

## Abstract

Aging and overweight increase the risk of developing type 2 diabetes mellitus. In this cross-sectional study, we aimed to investigate the potential mediating role of T-EMRA cells and inflammatory markers in the development of a decreased insulin sensitivity. A total of 134 healthy older volunteers were recruited (age 59.2 (_SD_ 5.6) years). T cell subpopulations were analyzed by flow cytometry. Furthermore, body composition, HOMA-IR, plasma tryptophan (Trp) metabolites, as well as cytokines and adipokines were determined. Using subgroup and covariance analyses, the influence of BMI on the parameters was evaluated. Moreover, correlation, multiple regression, and mediation analyses were performed. In the subgroup of participants with obesity, an increased proportion of CD8+EMRA cells and elevated concentrations of plasma kynurenine (KYN) were found compared to the lower-weight subgroups. Linear regression analysis revealed that an elevated HOMA-IR could be predicted by a higher proportion of CD8+EMRA cells and KYN levels. A mediation analysis showed a robust indirect effect of the Waist-to-hip ratio on HOMA-IR mediated by CD8+EMRA cells. Thus, the deleterious effects of abdominal obesity on glucose metabolism might be mediated by CD8+EMRA cells in the elderly. Longitudinal studies should validate this assumption and analyze the suitability of CD8+EMRA cells as early predictors of incipient prediabetes.

## 1. Introduction

During aging, the adaptive immune system undergoes a remodeling process characterized by altered T cell subtype composition and changes in major T cell functions. While the number of CD8+ cells generally increases, the proportion of CD4+ cells slightly decreases, resulting in a reduced CD4+/CD8+ T cell ratio [1]. Within both T cell populations, the relative proportion of cells with a naïve phenotype progressively decreases. Homeostatic proliferation of existing naive T cells in the periphery partially compensates for the reduced number of naive T cells [2]. Functionally, the T cell receptor (TCR) repertoire is diminished, resulting in a limited response against emerging pathogens [3]. Accompanying the exhaustion in the naïve T cell pool, an accumulation of terminally differentiated cells occurs. This is caused by repeated T cell stimulation, e.g., by latent viruses, such as cytomegalovirus (CMV) infection, or by a state of chronic inflammation [4,5].

T effector memory re-expressing CD45RA cells (T-EMRA cells) represent a heterogeneous group of terminally differentiated cells characterized by their surface expression profile CCR7−/CD45RA+. These cells can be further subdivided regarding their expression of CD57 into a functionally viable “young” fraction (CD57−) with proliferative and antiviral capacity and a senescent fraction (CD57+) with extensive functional quiescence that is proliferation incompetent in response to antigen-specific stimulation and susceptible to apoptosis upon T cell activation [6,7]. With age, the number of T-EMRA cells increases, which can be interpreted as a hallmark of immunosenescence [8]. T-EMRA cells are suggested to contribute to various pathological processes by exacerbating inflammation and exhibiting atypical cytotoxic activity towards endogenous structures, such as the vascular endothelium [9,10]. Moreover, T-EMRA cells have been identified as predictors of cardiovascular mortality in the elderly and as risk factors of graft dysfunction in kidney transplant recipients [11,12].

Accelerated T cell differentiation is bi-directionally related to the development of a chronic low-grade inflammation, termed “inflammaging” [13,14]. On the one hand, chronic inflammation is a driver of T cell differentiation by constantly activating existing immune cells. On the other hand, highly differentiated T-EMRA cells represent a pro-inflammatory phenotype and a source of inflammatory factors [10,15]. The concentration of various inflammatory cytokines, such as interleukin (IL)-1β, IL-6, and tumor necrosis factor-α (TNF-α), progressively increases during aging, which contributes to age-associated morbidity and mortality [16]. Overweight superimposed on aging accelerates low-grade inflammation, which might play a leading role not only in the pathogenesis of cardiovascular but also of metabolic diseases, such as diabetes type 2. Accordingly, patients with prediabetes or diabetes show significantly lower insulin sensitivity, increased proportions of terminally differentiated T cells, and higher levels of inflammatory cytokines compared to metabolically healthy individuals [17]. Besides glucose metabolism, the kynurenine (KYN) pathway as the major route of tryptophan degradation represents another popular example of the interplay between metabolism and inflammation. The initial and rate-limiting enzyme indoleamine 2-3-dioxygenase (IDO1) strongly increases in response to elevated proinflammatory cytokine levels [18], leading to a stimulation of the KYN pathway that has been demonstrated in several inflammation-associated pathologies [19,20].

Based on these findings, we speculate that T-EMRA cells, as well as parameters of the kynurenine pathway, may be related to a disturbance of insulin sensitivity in obesity and that this is detectable even in clinically healthy elderly subjects. We explicitly addressed the question: Do increased proportions of T-EMRA cells or activation of the KYN pathway favor an early dysregulation of insulin sensitivity in subjects aged 50–70 without any previous chronic illness? Furthermore, we asked whether T-EMRA cells and changes in Trp metabolites are potential mediators between abdominal obesity and disturbed insulin sensitivity.

## 2. Materials and Methods

The current work is based on a recent study of our group [21]. Blood samples were obtained from the same subjects. Data regarding anthropometric and physiological characteristics have been incorporated from this publication.

### 2.1. Study Participants

We enrolled 134 men and women from the general population in Hannover, Germany, between August 2018 and March 2019. Recruitment was conducted widely distributed via advertisements in local newspapers and public notices. The inclusion and exclusion criteria for each participant were determined using a formalized questionnaire. Inclusion criteria for participation were age ≥50 and ≤70 years, no regular exercise training aside from the daily activities for at least 2 years, and stable body weight (±5 kg) for at least 6 months. Exclusion criteria were defined as cardiovascular diseases (angina pectoris, myocardial infarction, stroke, peripheral arterial occlusive disease, heart failure, cardiac arrhythmia), type 1 and 2 diabetes, renal insufficiency and liver diseases, blood coagulation disorders, chronic gastrointestinal disorders (e.g., ulcers, Crohn’s disease, pancreatic insufficiency), immunological diseases (e.g., autoimmune diseases), intake of immunosuppressive drugs or laxatives, intake of supplements containing n3-FAs, smoking, alcohol, drug and/or medicine dependency, pregnancy or lactation, retraction of the consent by the subject, concurrent participation in another clinical study, and participation in a study in the last 30 days. Ethical approval was provided by the Ethics Commission of the Medical Chamber of Lower Saxony (Hannover, Germany). Following the guidelines of the Declaration of Helsinki, written informed consent was obtained from all participants before participation in the study. Study design and the procedure for the different analyses are shown in Figure 1a.

### 2.2. Body Weight and Body Composition

Waist and hip circumferences were measured using a measuring tape in a standing position. Waist-to-hip-ratio (WHR) was calculated. Height was measured with a stadiometer (seca GmbH & Co. KG, Hamburg, Germany) and body weight was measured on a digital scale to the nearest 0.1 kg (seca GmbH & Co. KG, Hamburg, Germany). BMI was calculated by the ratio of weight to the squared height. Body composition was analyzed to a nearest of 0.1 kg using a bioelectrical impedance analyzer (BIA) (Nutriguard M, Data Input Company, Darmstadt, Germany) and the software NutriPlus© 5.4.1 (Data Input Company, Darmstadt, Germany). For the measurements, the participants arrived rested and after an overnight fast (≥10 h). They were instructed to lay down on a stretcher and rest for at least five minutes to ensure a balanced distribution of body fluids before the measurement. During the measurement, participants were instructed to lay still and in a relaxed position with the limbs slightly bend from the torso. All measurements were carried out by the same study personnel.

### 2.3. Blood Sampling

Blood samples were taken in the morning following a resting period and after an overnight fast (≥10 h) using serum, EDTA, and NaF Glucose tubes (Sarstedt AG & Co. KG, Nümbrecht, Germany). Blood was either processed directly or stored at −80 °C in the form of serum and plasma for later analysis.

### 2.4. Analysis of Glucose Metabolism and Insulin Resistance

Analysis of markers of glucose metabolism was performed by a certified laboratory (Laborärztliche Arbeitsgemeinschaft für Diagnostik und Rationalisierung e.V., in Hannover, Germany). Fasting glucose was analyzed photometrically (Beckman Coulter GmbH, Krefeld, Germany). HbA1c was analyzed using high-performance liquid chromatography (HPLC) (Bio-Rad Laboratories GmbH, Feldkirchen, Germany). For the determination of insulin, the electrochemiluminescence immunoassay method (ECLIA) using cobas 801e (Roche Diagnostics GmbH, Mannheim, Germany) was applied. Insulin sensitivity was evaluated using the homeostatic model assessment (HOMA): HOMA-IR = fasting insulin (µU/mL) × fasting blood glucose (mg/dL)/405 [22].

### 2.5. Analysis of T Cell Subpopulations

Peripheral Blood Mononuclear Cells (PBMCs) were isolated from fresh EDTA whole blood by ficoll density gradient centrifugation. After PBMCs were washed, 1 × 10^6^ cells in 100 µL PBS were stained for 20 min in the dark with 5 µL of different fluorescence-coupled antibodies, respectively (BioLegend Inc., San Diego, CA, USA & ImmunoTools GmbH, Hamburg, Germany). The antibody cocktails were composed as follows. Analysis of CD4+ subtypes: anti-CD4-FITC (clone MEM-241), anti-CD197(CCR7)-PE (clone G043H7), anti-CD45RA-PerCP (clone HI100); Analysis of CD8+ subtypes: anti-CD8-FITC (clone MEM-31), anti-CD197(CCR7)-PE (clone G043H7), anti-CD45RA-PerCP (clone HI100). Percentages of naïve (CD45RA+/CCR7+), central memory (CD45RA-/CCR7+), effector memory (CD45RA-/CCR7−), and EMRA (CD45RA+/CCR7−) CD4+ and CD8+ T cells were quantified by flow cytometer FC 500 using the CXP software (Beckman Coulter, Irving, TX, USA). The gating strategy is shown in Figure 1b and was implemented according to Koch et al. [8]. Besides, CD4+/CD8+ T cell ratio was determined. 

### 2.6. Analysis of Tryptophan Metabolites

Trp and its metabolites KYN, quinolinic acid (QA), and kynurenic acid (KA) were measured via high-performance liquid chromatography (HPLC) coupled to a mass spectrometer (MS). Serum was stored in 50 µL aliquots at −80 °C until analysis. The analysis was performed on a Waters ACQUITY UPLC^®^ system equipped with an ACQUITY UPLC^®^ HSS T3 analytical column coupled to a Xevo^®^ TQ-XS triple quadrupole mass spectrometer (Waters, Eschborn, Germany) as described elsewhere [20].

### 2.7. Analysis of Plasma Cytokines

Plasma levels of IL-1β, IL1ra, IL-6, IL-15, TNF-α, CX3CL1/fractalkine, adiponectin, and resistin were determined using a human Magnetic Luminex Assay (Bio-Techne, Abingdon, Oxon, UK) and a Magpix Luminex instrument (Luminex Corp, Austin, TX, USA) according to the manufacturer’s instructions.

### 2.8. Analysis of Cytomegalovirus (CMV) Serostatus

Serum anti-CMV immunoglobulin G (IgG) antibodies were detected using a semiquantitative sandwich enzyme-linked immunosorbent assay (ELISA-Viditest anti-CMV IgG, VIDIA, Czech Republic). The procedure followed the manufacturer’s instructions. End-point optical density was measured by the Emax Plus ELISA reader (Molecular Devices, Sunnyvale, CA, USA).

### 2.9. Statistical Analysis

First, descriptive statistics were performed for all measured variables and results are given below as mean ± SD. In the next step, we tested all observed variables regarding their distribution features. As most of them did not meet the criterion of a normal distribution, we used Spearman’s rank correlation to determine possible relationships between measurements of body composition (as indicated by BMI and WHR), metabolism (as indicated by HOMA-IR and Trp metabolites), CD4+ and CD8+EMRA cells, as well as pro-inflammatory cytokines (e.g., IL-6, TNF-α) and adipokines (adinopectine, resistin) for exploarory purposes. Missing data were addressed using pairwise deletion.

As a preliminary analysis, we further divided our study collective into three BMI subgroups: group 1: 18.5–24.9 kg/m^2^ (normal weight); group 2: 25–29.9 kg/m^2^ (overweight); group 3: >30 kg/m^2^ (obesity). We calculated an ANCOVA for the dependent variable CD8+EMRA cells and Trp metabolite KYN and the independent variable BMI group using age as a covariate to assess the impact of a BMI classified within the obese range (>30) on the proportion of T-EMRA cells and enhanced Trp metabolites.

In the next step, we analyzed the impact of obesity, proportion of T-EMRA cells, and levels of Trp metabolites as potential predictors of abnormal glucose metabolism (indicated by HOMA-IR) using multiple regression analyses. We added several product terms to the model to test for an interaction between abdominal obesity, immune aging, and Trp metabolites. Before multiple regression analysis, we z-standardized all variables. To identify a possible mechanism that underlies the observed relationship between abdominal obesity and disturbed glucose metabolism, we tested the mediation of this relationship by assessing the factors of immune aging, as well as the concentration of tryptophan metabolites. More precisely, we calculated a mediation analysis using bootstrapping with 5000 bootstraps with the predictor variable WHR, the mediators CD8+EMRA cells as well as KYN, and HOMA-IR as the outcome variable. For all statistical analyses as well as data visualization, we used JASP, version 14.0. *p* values < 0.05 are considered significant.

## 3. Results

### 3.1. Baseline Characteristics

After screening for eligibility, 134 participants met the eligibly criteria (Figure 2). Their data are specified as mean ± SD. Of all participants, 72% were female and 28% were male with a mean age of 59.2 ± 5.6 years. Participants were further characterized by a weight of 83.0 ± 20.3 kg, BMI of 28.3 ± 5.8 kg/m^2^, a WHR of 0.85 ± 0.09, and a HOMA-IR of 2.72 ± 2.26. The study population can be classified as healthy, but pre-obese. All measured baseline characteristics are summarized in Table 1.

### 3.2. Associations between Body Composition, Glucose Metabolism, T-EMRA Cells, Trp Metabolites, and Cytokine Status

Proportions of T cell subpopulations and levels of measured cytokines, adipokines, and Trp metabolites of all participants can be found in Appendix A.

We examined the Spearman rank correlation between either BMI, WHR, HOMA-IR, CD8+EMRA cells, IL-6, resistin, and KYN separately for each participant. Note that for all upcoming results, we used the z-standardized variables. The correlation analysis revealed correlations between WHR and CD8+EMRA cells (r = 0.351, *p* < 0.001), IL-6 (r = 0.249, *p* = 0.021), resistin (r = 0.221, *p* = 0.046), KYN (r = 0.207, *p* = 0.041), as well as HOMA-IR (r = 0.447, *p* < 0.001). CD8+EMRA cells were correlated with KYN (r = 0.242, *p* < 0.05), BMI (r = 0.235, *p* < 0.05), and HOMA-IR (r = 0.250, *p* < 0.05). IL-6 was associated with BMI (r = 0.352, *p* < 0.001). Resistin was positively correlated with BMI (r = 0.349, *p* < 0.01). Furthermore, we found a correlation between KYN and BMI (r = 0.355, *p* < 0.001) and HOMA-IR (r = 0.224, *p* < 0.05). All results of the correlation analysis are depicted in Figure 3.

### 3.3. Effect of BMI on T-EMRA Cells and Trp Metabolism

As a pre-analysis, we tested whether a BMI classified within the obese range (>30) in particular leads to an increased proportion of T-EMRA cells as well as to enhanced levels of Trp metabolites. The calculated ANCOVAs including age as a covariate revealed a significant main effect of BMI group on the number of CD8+EMRA cells, F(2, 89) = 3.897, *p* = 0.024, η*p*2 = 0.081, as well as on the level of the Trp metabolite KYN, F(2, 94) = 6.625, *p* = 0.002, η*p*2 = 0.124. Post-hoc analyses using Bootstrapping indicated that, overall, a BMI > 30 is likely to lead to an increased proportion of CD8+EMRA cells (BMI group 2 vs. 3: *p* < 0.05) as well as to enhanced levels of KYN (BMI group 1 vs. 3 and BMI group 2 vs. 3: *p* < 0.05). Results are depicted in Figure 4a.

Further analyses revealed no main effect of sex, muscle mass, or weekly physical activity level on CD8+EMRA cells and KYN levels. Moreover, no significant interaction effects, for example, between sex and BMI, were found (data not shown).

### 3.4. Moderation of the Glucose Metabolism by Abdominal Obesity, the Proportion of T-EMRA Cells, and Trp Metabolites

We performed a multiple regression analysis investigating whether WHR, the proportion of T-EMRA cells, and changes of Trp metabolites are predictors of an altered glucose metabolism. Here, the WHR was used because it represents a better indicator of abdominal obesity and related health risks than BMI [23]. Furthermore, we tested whether the proportion of T-EMRA cells and KYN levels moderate the effect of abdominal obesity on the disturbances of glucose metabolism. The performed linear regression analysis revealed that the HOMA-IR could be predicted by the proportion of CD8+EMRA cells and KYN. Results showed that the HOMA-IR increases significantly when CD8+EMRA cells increase (β = 0.302, *p* < 0.01). Moreover, increased KYN levels correlated significantly with increases in HOMA-IR (β = 0.212, *p* < 0.05) (Figure 4b). The total variance explained by the model as a whole was adjusted R2 = 0.232, F(3, 89) = 10.271, *p* < 0.001. The model that included the product terms did explain little additional variance in the HOMA-IR score, adjusted R2 = 0.276, F(7, 85) = 6.001, *p* < 0.001. Results revealed a significant interaction between the proportion of CD8+EMRA cells and KYN (β = 0.277, *p* < 0.01) reflecting that higher CD8+EMRA cells accompanied by higher KYN levels lead to a stronger increase of HOMA-IR (Figure 4c).

### 3.5. Abdominal Obesity Affects T-EMRA Cells and Trp Metabolites, Which in Turn Influences HOMA-IR

We further analyzed whether the effect of abdominal obesity on glucose metabolism is mediated by immune senescence and Trp metabolites. A mediation analysis using bootstrapping showed a robust indirect effect of WHR on HOMA-IR mediated by CD8+EMRA cells (β = 0.084, *p* < 0.05). Additionally, a mediating tendency for KYN (β = 0.058, *p* = 0.077) was observed. Furthermore, the analysis revealed a significant covariation between KYN and CD8+EMRA cells (β = 0.22, *p* < 0.05) (Figure 4d).

## 4. Discussion

Effects of an individual’s body composition, namely an increased BMI or abdominal obesity, are well-documented drivers of a dysregulated glucose metabolism [24]. However, potential mediators of this effect remain largely unclear, especially in a population of healthy aged subjects. Our findings show statistical associations between glucose metabolism, levels of Trp metabolites, and proportions of circulating T-EMRA cells. In particular, increased numbers of CD8+EMRA cells and increased levels of KYN could be potential predictors of a dysregulated glucose metabolism, even in clinically as yet unremarkable individuals. The present data also indicated that both the proportion of CD8+EMRA cells and KYN levels might play a mediating role between the increase in abdominal fat and an elevated HOMA-IR. This was supported by the fact that, in particular, the group of subjects with BMI classified as obese had increased KYN levels and CD8+EMRA concentrations.

A methodological limitation of our study in the context of flow cytometric evaluation was that we gated the CD4+ and CD8+ cells separately and not on one plot. CD4+ CD8+ double-positive cells could thus have been captured twice [25]. Although there usually is only a small proportion of double-positive cells, this aspect should be considered in future work. Due to the cross-sectional design, the present study can provide only initial insights into possible relationships but no causal conclusions can yet be drawn with absolute certainty. Nevertheless, the strength of the present work is the large representative human sample in an age-cohort of interest concerning the first occurrence of lifestyle-related diseases. Thus, high generalizability of our results can be assumed. Besides, our preliminary considerations were based on knowledge of biological correlations and mechanisms in experimental (animal) studies, as described below. This legitimizes our statistical approach, supports our tentative conclusions, and reinforces the utility of follow-up studies in a longitudinal design to validate the assumptions we made here.

### 4.1. The Role of T-EMRA Cells in Obesity-Related Disturbance of Glucose Homeostasis

Overweight and obesity facilitate the incidence of many internal diseases and various data suggest that this occurs in parallel to the age-related remodeling of the adaptive immune system, particularly due to the accumulation of terminally differentiated T cells [26]. An important driver of these changes are lifestyle factors, such as inactivity and malnutrition, which lead to an increase in visceral adipose tissue (VAT). On one hand, the expansion of VAT favors a dysfunctional metabolic environment, but on the other hand, it accelerates T cell differentiation [27,28]. In this regard, it was previously shown that obesity accelerates thymic involution resulting in a lower naïve T cell production, leading to a proportional increase of terminally differentiated T cells [29]. In parallel, the expansion of visceral adipose tissue accelerates T cell differentiation by the progressive release of pro-inflammatory adipokines and cytokines [30]. A self-reinforcing cycle of progressive T cell differentiation and inflammatory cytokine production occurs because T-EMRA cells themselves secrete a variety of pro-inflammatory factors [10,15].

Our results concretize these data by showing an increased accumulation of T-EMRA cells in aged subjects with obesity and the association between abdominal fat and proportions of CD8+EMRA cells. As described in the introduction, T-EMRA cells are a heterogeneous subset of cells consisting of functional (CD57−) and senescent (CD57+) cells which occur in approximately equal proportions [6]. This further subdivision was not performed in the context of our study. However, results of mouse experiments suggest that mainly the exhausted, senescent T-EMRA cells may be crucial mediators of obesity-associated dysregulated glucose homeostasis.

Yi et al. [31] adoptively transferred senescent CD8+ T cells isolated from the spleens of mice fed a high-fat diet into young mice. Subsequently, the recipients showed aggravation of systemic glucose tolerance and insulin sensitivity compared to control mice. Another causal relationship between T cell aging and metabolic dysfunction was indicated in a study with obese mice. Here, the deletion of senescent cells alleviates the high-fat diet-induced metabolic dysfunction [32]. Cell culture experiments demonstrated a direct interaction between aged senescent CD8+ cells and hepatic insulin sensitivity by suppressing the activity of glycolytic enzymes [31]. However, the detailed molecular interactions between CD8+EMRA cells and metabolic pathways are still unknown. We can only speculate that cellular interactions or secreted molecules reduce tissue insulin sensitivity and possibly disrupt β-cell function. In this context, several inflammatory cytokines, such as IL-6, TNF-α, adiponectin, and resistin, are directly or indirectly involved in the regulation of insulin sensitivity [33,34]. Moreover, IL-6 and resistin have been associated with abdominal obesity on the one hand and an increased number of senescent T cells on the other hand [35]. For T-EMRA cells in general, secretion of TNF-alpha has also been demonstrated [10]. However, in our study, we did not observe a mediating role of any of these cytokines at the systemic level. Thus, follow-up studies should erode this potential mechanism at the tissue or cellular level.

Our finding that CD8+EMRA cells might represent a physiological mediator of disturbed glucose homeostasis is important from two perspectives. First, it would be important to investigate the potential direct molecular interactions, how these immune cells interact with beta cells or insulin-sensing organs, to uncover conceivable preventive and therapeutic targets. On the other hand, analyses of CD8+EMRA cells could represent an early predictive marker of a metabolic shift before subjects show clinically relevant symptoms of metabolic diseases. These assumptions need to be addressed in future (longitudinal) studies. In this context, a more specific subdivision of the T-EMRA cell population concerning additional surface markers, e.g., senescence marker CD57, or functional properties would be beneficial. This could generate further mechanistic associations and possibly increase the uniqueness of the observed effects.

### 4.2. KYN as a Mediator between Abdominal Fat and a Disturbed Glucose Metabolism

Our results show associations between Trp metabolite KYN and disturbed glucose metabolism. This finding corresponds with previous studies which found the involvement of tryptophan metabolites in the control of pancreatic hormonal secretion and hepatic glucose production [36]. Increased KYN levels suggest increased activity of IDO1. The turnover rate of this enzyme is dependent on inflammatory signaling. In our work, we could not find a strong correlation between KYN levels and any of the detected cytokines at the systemic level. However, based on our data, we cannot draw any conclusion about possible molecular interactions at the tissue level, for example, in the abdominal adipose store. Enhanced KYN levels in the subgroup with obesity and the association with WHR suggests that there is a mechanistic link to the amount of abdominal fat. Previous studies on this topic found that IDO1 expression and activity are enhanced in adipose tissue during conditions of inflammation [37]. Since chronic inflammation is also favoring T cell differentiation and metabolic disturbances related to overweight, the inflammatory shift in adipose tissue seems to be a precursor for early pathological processes [38]. The positive association between KYN and HOMA-IR corresponds with previous data which found an association between Trp metabolites and insulin resistance. Mechanistically, Trp metabolites have been shown to affect the biosynthesis, release, and activity of insulin [39].

## 5. Conclusions

Overweight facilitates the incidence of many internal diseases, specifically in those subjects at the end of the BMI spectrum. Especially metabolic pathologies develop slowly and gradually, and such changes can be detected even in individuals clinically assessed as healthy, like in the present study [40].

The increase in body fat in elderly subjects is related to increased CD8+EMRA populations as well as elevated plasma KYN levels and both are in turn associated with pre-clinical metabolic dysregulation. These findings integrate previous data of patients in different disease groups and are more remarkable because we found it in a population of clinically healthy older subjects. It is assumed, that the WHR dependent metabolic changes represent a preliminary stage of incipient pathological processes, which announce themselves especially in the subjects who develop obesity. The study once again highlights the importance of maintaining reliable control over the development of abdominal fat through specific lifestyle factors, especially in older age. At the same time, the data suggest that changes in Trp metabolites and an increase in CD8+EMRA cells may represent early predictors of prediabetes. This needs to be confirmed in longitudinal studies. Furthermore, from a basic science perspective, the interaction of CD8+EMRA cells with metabolic changes related to glucose regulation and Trp metabolites is of particular interest.

## Figures and Tables

**Figure 1 cells-10-00998-f001:**
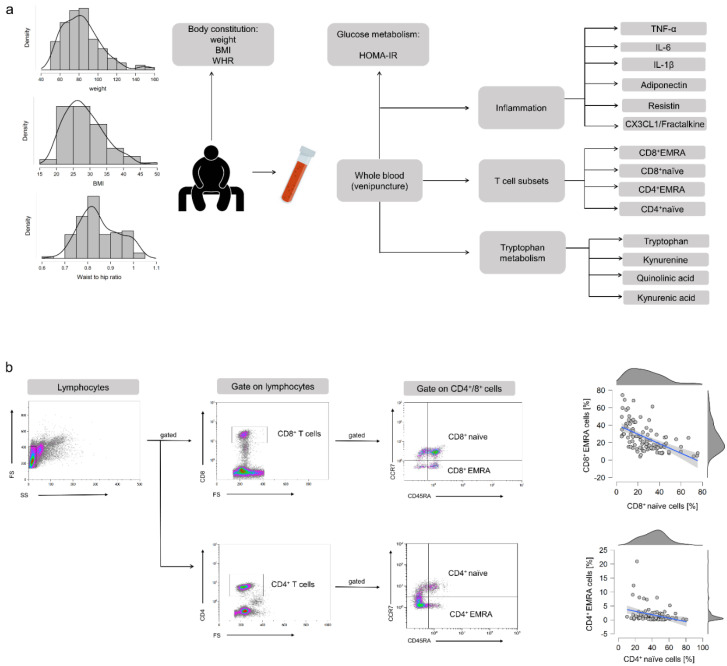
(**a**) Presentation of the study design and the course of analyses. (**b**) Flow cytometric analysis of T cell subpopulations. Lymphocytes were gated using the forward- and sideward scatter. Populations were subdivided into CD4+ T-helper cells and CD8+ cytotoxic T cells. Both cell types were further differentiated into naïve (CD45RA+/CCR7+), central memory (CD45RA−/CCR7+), effector memory (CD45RA−/CCR7−), and effector memory re-expressing CD45RA (CD45RA+/CCR7−) cells. Associations between naïve and EMRA T cells.

**Figure 2 cells-10-00998-f002:**
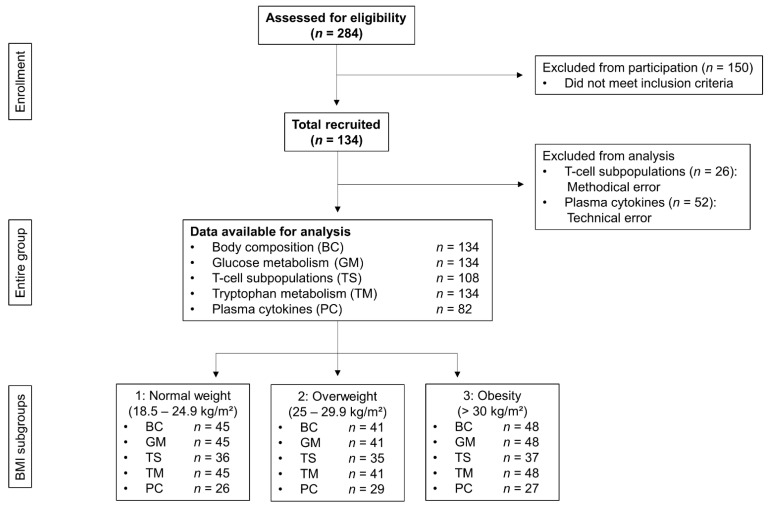
Flow chart of all participants screened and analyzed.

**Figure 3 cells-10-00998-f003:**
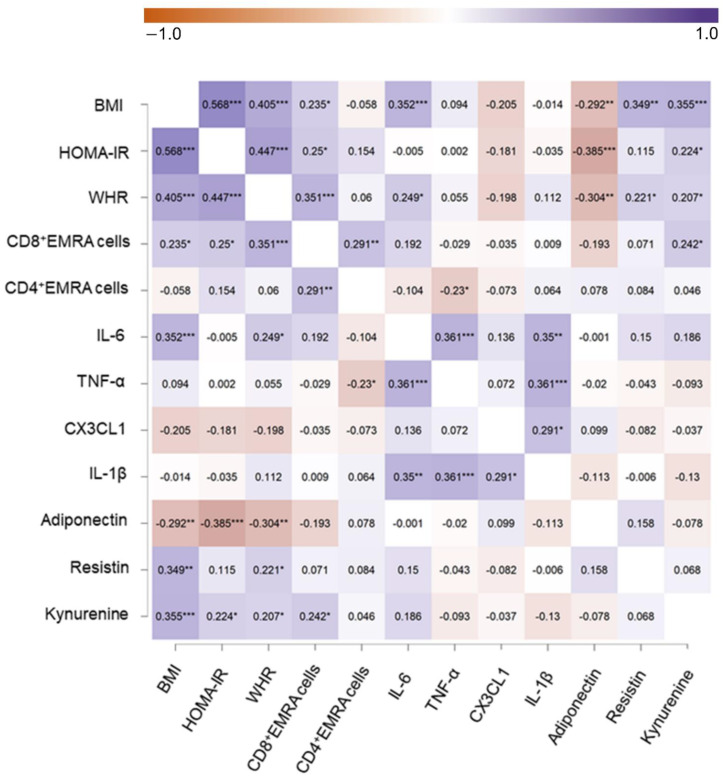
Correlation heat map including BMI, HOMA-IR, WHR, CD8+ and CD4+EMRA cells, IL-6, TNF-α, CX3CL1, adinopectin, resistin, and KYN. Raw *p* values of Spearman’s rank s correlations are presented. * means *p* < 0.05, ** means *p* < 0.01, *** means *p* < 0.001 not corrected for multiple tests.

**Figure 4 cells-10-00998-f004:**
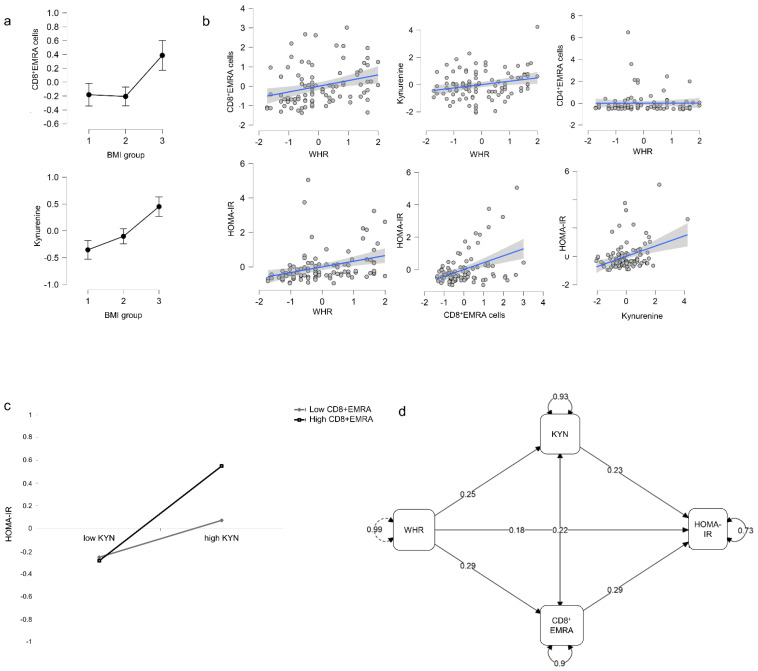
(**a**) Proportion of CD8+EMRA cells and KYN in BMI group 1 (BMI 18.5–24.9 kg/m^2^), group 2 (BMI 25–29.9 kg/m^2^), and group 3 (BMI > 30 kg/m^2^). (**b**) Correlations between WHR and the proportion of CD8+EMRA cells, WHR and KYN, WHR and proportion of CD4+EMRA cells, WHR and HOMA-IR, CD8+EMRA and HOMA-IR, as well as KYN and HOMA-IR. (**c**) Associations between low/high kynurenine levels, levels of CD8+EMRA cells, and HOMA-IR. (**d**) Interaction of WHR on HOMA-IR mediated by CD8+EMRA cells as well as by KYN. Furthermore, a significant covariation between KYN and CD8+EMRA is demonstrated.

**Table 1 cells-10-00998-t001:** Baseline characteristics (mean ± SD) of all participants.

Baseline Characteristics	
Sex [f/m]	[96/38]
Age [years]	59.2 ± 5.6
Height [m]	170.9 ± 8.7
Body weight [kg]	83.0 ± 20.3
BMI [kg/m^2^]	28.3 ± 5.8
Waist circumference [cm]	93.30 ± 14.53
Hip circumference [cm]	108.32 ± 12.15
WHR	0.85 ± 0.09
Fasting Glucose [mg/dL]	93.13 ± 16.78
HbA1c [%]	5.44 ± 0.45
Insulin [µU/mL]	11.39 ± 7.96
HOMA-Index	2.72 ± 2.26
CMV positive [Yes/no]	[72/63]

## Data Availability

The data generated during the current study are available from the corresponding author on reasonable request.

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
