# Peer review of "Abdominal Obesity-Related Disturbance of Insulin Sensitivity Is Associated with CD8+ EMRA Cells in the Elderly"

_cells, 2021, doi:10.3390/cells10050998_

Round 1

Reviewer 1 Report

Diabetes mellitus is a frequent pathology and its prevalence is on the rise so well-designed studies on this particular aspect are imperative. Furthermore, it touches on the issue of inflammaging, another compelling subject matter.

This is a well-constructed and thorough study, on an interesting topic, however I have a few remarks.

Did the authors take into consideration smoking status?

What was the reasoning behind the chosen cytokines?

Did gender affect in any way the results?oo

As a future research path, it would be interesting to explore the role of oxidative stress.

Reviewer 2 Report

Dear authors, thank you for submitting this interesting report in such an interesting human cohort. Below are specific comments and questions surrounding this manuscript. I hope these are of use to improve the potential impact of this interesting work.

Abstract

Line 24: mean given to 1 d.p but SD given to 2 d.p – needs to be consistent We also don’t know the units being presented here (SD or SEM?)

It would be of benefit to perhaps detail the method (Flow cytometry) used for cell identification

Introduction

Lines 62-63: It is perhaps misleading to call all T-EMRA cells senescent as these merely represent T cells terminally differentiated into an effector memory phenotype, which are still functionally viable and able to fight pathogens as part of a healthy immune response rather than just being dysfunctional (senescent) cells (PMID: 28481945), whilst other molecular drivers are required for them to become truly senescence (PMID: 29024417).

Methods

It might be worth specifying how inclusion an exclusion criteria were identified for each participant (i.e., by self – report, verbally, or by a formalised questionnaire/ health screen).

Figure 1b. The gating strategy included here does not show how regulatory T cells (detailed in section 2.5 as being measured) were identified. All cell types that data are presented for should be included in the representative flow cytometry gating tree. Further, the conventional approach to gating CD8 and CD4 T cells is to have CD4 against CD8 on one plot, wherein there will be a small portion of double-positive CD4+CD8+ T cells (See: PMID: 15110234), which are likely captured twice when gating CD4 and CD8 separately. Furthermore, it is standard practice to gate out doublets prior to cell phenotyping in flow cytometry so as to avoid clumped cells (i.e., two cells passing through the fluidics together; See: PMID: 29023707 and 31633216). If this has not been conducted then it should be reanalysed accordingly to ensure accuracy in the final results. Finally, in the T cell subset plots, the CD4+ CCR7/CD45RA representative plot looks as though there is considerable spectral overlap between these channels causing a curved transition between NA, CM, and EM subsets. It may be that each of these populations aren’t adequately captured should this be a compensation issue that is apparent in all of the raw flow cytometry data. Were FMO/isotype controls used to inform gating strategies for the T cell subsets? Given this is the specific study focus this is a point of major consideration to address.

Section 2.3: Were bloods sampled immediately upon arrival or following a resting period (as detailed in section 2.2?) This should be specified in the text.

Line 143: Is ‘insulin resistance’ the correct terminology here? Should it not be more accurately described as a measure of insulin sensitivity given insulin resistance is a more clinical set point?

Section 2.5: The specific antibody cocktails and their fluorochromes need to be specifically detailed here in full. Ideally antibody clone information should also be specified in this section. (e.g., anti-CD4–FITC, clone GK1.5).

Line 156: Treg cells have been specified as CD3+CD4+CD25+, however, this is likely not adequate enough a gating strategy to truly call these Treg cells. Ideally the transcription factor FoxP3 would be used to identify CD25+FoxP3+ T reg cells. Indeed, especially in the present study where effector memory cells are being addressed, it is known that CD25 is a marker of activation on effector T cells (being the IL2 receptor α chain) and therefore presenting these cells as Tregs may derive misleading findings should there be proliferative/activation responses in other non-Treg T cells as has been reported in lean vs overweight/obese humans (PMID: 25388403). Therefore, without the inclusion of FoxP3 in the flow cytometry panels Treg cell data should ideally be omitted.

Section 2.8: excellent control in the statistical approaches described in Lines 177-185. However, the means of displaying data are not detailed (SD/SEM etc) and require detailing.

Line 207: Age is presented to 1 d.p for the group mean but to 2 d.p for the SD/SEM (Please specify which of these is being presented for clarity). Both should be to 1 d.p.

Line 208-209: The same point about decimal place consistency for the BMI data. Please make both the mean and SD/SEM decimal places consistent.

Figure 2: No details about how participants were recruited is included here (referral/ local advertisement etc). This should be included before its initial mention on Lines 299-300.

Table 1: Decimal places for each row (across Mean and SD) have to me made the same. Between variables this can be varied between one or two decimal places but not within the same variable.

Was there a bias in the sex-split between young and old? Or between the different BMI groups? Or across the age-span? This is a very important consideration should males or females be overly represented in one cohort compared to another allowing biological sex differences to influence the results and interpretation thereof (See: PBMI: 32029736).

Figure 3: In the text accompanying this figure, and in the methods section, it says a Spearman’s ρ was used to compute correlation analyses due to many variables not being normally distributed but within the legend to figure 3 it says pearson’s correlations are presented. Which of these was used and can this be made consistent across the manuscript.

Are P values designated within Figure 3 the post-bonferroni corrected values or raw? This must be included in the figure legend.

Was any account for physical activity or muscle mass taken in this analysis given their potent role in glucose homeostasis? If not, this requires consideration in the discussion.

A colour coded key for the heatmap scale should be included as standard practice.

Lines 291-293: Although sophisticated modelling was conducted, it is unlikely that this cross-sectional characterisation snap shot of these individuals, without lifestyle measurements etc, is able to say with such confidence that EMRA and KYN are playing a mediating role between abdominal fat and HOMAIR. We cannot know the series of events, the expected, respective magnitudes of change for each of these variables, or their interplay from such a study design. We must remain tentative suggesting what results mean rather than firmly stating cause and effect. Indeed, given the role of the adipose immune system in particular on glucose metabolism, inflammation, and TPY-KYN metabolism (PMID: 23625535, 32351500) this is a highly complex system that the present results can offer insights towards potential links but not firm conclusive links. If the wording of the discussion be adjusted to become more conservative in tone then this would certainly increase the impact of these findings for potential readers who may otherwise have reservations about concrete statements of mechanisms.

This is perhaps best exemplified by the tone taken in Lines 301-305 where the exact value of the present data are very nicely summarised. If this tone could be adopted throughout it would certainly improve the manuscript readability. Indeed, the sentence across lines 303-305 is an excellent summary of the value of these results to the literature. The fact that these results are in humans is a true benefit of the study, whereas animal work would be required to confirm mechanisms. This is in no way, however, to detract from the present result in such a large cohort of humans.

Lines 353-355: This statement is perhaps somewhat misleading as only measurements at the systemic level were made and therefore the interaction between cytokines and KYN within target tissues cannot be discerned. This statement should ideally reflect the limitation of the level of measurement, leaving open the potential of such interactions occurring within tissues rather than purely ascribing wholly different signalling mechanisms. This is especially important when considering obese and older participants for whom a major adaptation is an enlarged abdominal adipose store, which may be a major contributing factor to this relationship which cannot be captured in the present analysis at the molecular level.

Lines 370-371: As with the comment above, a less concrete and more conservative phrasing should be used here to appropriately reflect the cross-sectional observational data captured here. It cannot be said with such certainty that these interactions are occurring in this direction.

Of particular note, however, to conclude this review is the topic of focus based on the title; T cell senescence. This is specified in the opening to this article but features nowhere in the conclusions of these findings, which is a major point of consideration. Moreover, when reading the title the reader expects to see readouts of specific T cell senescence, either by ex vivo functional analysis, or by the inclusion of cell surface markers of senescence. However, nowhere in the manuscript are there any phenotypic identifiers of T cell senescence and it is not enough to call T-EMRA cells purely senescent (as detailed above in this review). This requires consideration and major corrections to appropriately caveat these results appropriately.

Round 2

Reviewer 2 Report

Dear authors, Thank you for your commendable attempts to address this reviewers comments so thoroughly.

Title:

Thank you for amending the title – this much more appropriately reflect the focus of the study. Maybe qualifying very briefly what the association represents between glucose homeostasis and TEMRAs is might also further improve the title. I wonder also whether you need to clarify abdominal adiposity rather than obesity per se. I say this because the title originally mentions obesity and then ends ‘in healthy aged subjects’. It feels as though it jumps from cohort to cohort and you couldn’t describe (as is done in the abstract) subjects as healthy aged if also referring to abdominal obesity. Any alternative approaches that disentangles and distinguishes the use of obese and healthy older participants would be appropriate.

Introduction:

I would like to commend the authors on their restructuring of the introduction to this manuscript. The well-balanced arguments are equally well presented and the appropriate discussion around T-EMRA cells and senescence is now available for the reader.

Lines 88-90: the authors say ‘…this is detectable even in healthy elderly subjects.’ However, as part of this sentence was initially referring to the context of obesity, qualifying this specifically would be beneficial earlier in the sentence. Even just by saying ‘…a disturbance of glucose homeostasis in obesity, and that this is detectable even in healthy elderly subjects.’

Point 4: I am not sure whether line numbers are affected by the text that is crossed out but still remains because lines 98-99 in my manuscript are what must have been removed text.

Point 5: Thank you for addressing these extensive comments made in the first round of review. Thank you for your explanation surrounding the CD4/8 gating. I have followed the citations used in this manuscript that informed the flow cytometry gating profiles used here. You are correct in stating that the double positive population, biologically, is very small and unlikely to interfere considerably with results, though it is always worth visually checking no spectral overlap exists between the CD4 and CD8 parameters. Based upon your comments regarding the checking of the population of double positives, it appears that the authors were still able to identify clear CD4 and CD8 positive clusters when gated against one another. Therefore, I trust the approach used by the authors here and am satisfied to leave the flow gating profile example as originally laid out. Thank you for taking the time to discuss this topic. Regarding the inclusion of the methodological consideration around this topic, in my version of the manuscript it appears the final section of the manuscript conclusion is a description of the methodological problem with the flow gating. I first want to commend the authors for this comment which will benefit any readers’ concerns, but would this not better be suited to somewhere else within the discussion, leaving the reader with a summary sentence on the impact/relevance of the manuscript findings (as is currently presented in the preceding paragraph of the conclusion? It seems an unconventional place to place this methodological discussion, and feels unconventional to leave the reader at the final sentences with a method discussion rather than a punchy summary of the manuscript itself – which was the case in the original manuscript. Was this placement perhaps just a mistake and was meant to have been placed elsewhere?

Finally, regarding the compensation of data. Are all data for CD45RA against CCR7 as presented in the representative plots? It is much more usual to see this gating split as four quite distinct (which become less distinct clusters, blending together with increasing cell numbers) populations, which are at least free of ‘bends’ in the distribution, as is apparent in your representative plot. If all data weren’t like this representative plot, and more akin to the example linked here (taken from: https://www.thermofisher.com/antibody/product/CD197-CCR7-Antibody-clone-3D12-Monoclonal/35-1979-42), then this representative data should perhaps be exchanged for one that will cause less concern for readers. Alternatively, if FMOs were used, and the representative plots were indeed what was seen across all samples, then including FMO controls as a supplementary material would definitely be reassuring to readers who might at first be cautious when interpreting your results. If so, then presenting these controls is a very valuable addition to this manuscript. Of note, this does only seem to be an issue for the representative gating of CD4 T cells, as CD8 T cells looks much more like the example below.

Point 7: Thank you for clarification surrounding this amendment, and discussion on its context relative to HOMA-IR. I fully agree with your explanation here and am satisfied by the amendments.

Point 15: Thank you for such a detailed response with excellent statistical support for your argument. These are excellent analyses and show that biological sex did not unduly impact your findings, or skew/blur any results, statistically. Would the authors agree to include a single line to reference this finding, to also justify the work conducted here as a check, and to reassure sceptical readers who may want to know whether biological sex had an influence. Perhaps just one line to say that ‘no effect of biological sex was found across [parameters/broad terms] (data not shown).’ This isn’t a formal requirement, merely a suggestion that the authors are free to omit.

Point 18: As with the discussion around biological sex influence, as muscle mass and activity levels may be reasonably expected to be important potential cofounders for these types of data, would the discussion or relevant section of the results benefit from a single sentence to say these potential influencers were considered and were investigated but not found to statistically significantly influence the outcomes?

Point 22: I have made reference to the title at the top of this review. First of all, I very much would like to commend the work and effort that has gone into this edit, and the huge benefit this has had. The manuscript is now much more appropriately worded, with reference to its discussion of immune ageing and senescence.

Where possible, I have tried to follow the line references made throughout, but this has been very difficult as all tracked changes have been included within the text (including insertions and deletion), which, if presented to the reviewer in Microsoft word would be very easy to follow. This is likely not at all the authors fault, and merely reflecting the journals decision to format the reviewed manuscripts in its journal format as a pdf. Therefore, if any specific aspects have been overlooked, or confused for discussions elsewhere in the text, I would like to apologise for this, but stress how difficult it was to follow line references helpfully made by the authors in their responses given the journal formatting issues.

I have one specific in-text comment regarding the wording of the manuscript:

Line 301-302: check tense, were should read ‘are’, when referring to figure 3.
